## Neglected Tropical Diseases

# Navigating the cholera elimination roadmap in Zambia – A scoping review (2013–2024)

Nyuma Mbewe[1]*, John Tembo[2], Mpanga Kasonde[1‡], Kelvin Mwangilwa[1‡], Paul Msanzya Zulu[1‡], Joseph Adive Sereki[3‡], William Ngosa[1‡], Kennedy Lishipmi[4‡], Lloyd Mulenga[4‡], Roma Chilengi[1], Nathan Kapata[1], Martin Peter Grobusch[5]

1 National Cholera Elimination Taskforce, Zambia National Public Health Institute, Lusaka, Zambia, 2 HERPEZ Zambia – Institute for Infectious Disease Research, Lusaka, Zambia, 3 Regional Cholera Support Coordinator, International Federation of Red Cross Society, Lusaka, Zambia, 4 Ministry of Health Headquarters, Ndeke House, Lusaka, Zambia, 5 Department of Infectious Diseases, Center of Tropical Medicine and Travel Medicine, Amsterdam University Medical Centres, Location AMC, Amsterdam Infection and Immunity, Amsterdam Public Health, University of Amsterdam, Amsterdam, The Netherlands

☉ These authors contributed equally to this work.
‡ MK, KM, PMZ, JAS, WN, KL and LM also contributed equally to this work.
* nymbewe@gmail.com

## Abstract

### Background

Cholera outbreaks are increasing in frequency and severity, particularly in Sub-Saharan Africa. Zambia, committed to ending cholera by 2025, instead experienced its most significant outbreak in 2024. This review examines the perceived regression in elimination efforts by addressing two questions: (i) What is known about cholera in Zambia? and (ii) What are the main suggested mechanisms and strategies to further elimination efforts in the region?.

### Methodology/principal findings

A scoping literature search was conducted in PUBMED to identify relevant qualitative and quantitative research studies published between 1st January 2013 and 30th June 2024 using the search terms 'cholera' and 'Zambia'. We identified 53 relevant publications. With the increasing influence of climate change, population growth, and rural-urban migration, further increases in outbreak frequency and magnitude are expected. Risk factors for recurrent outbreaks, including poor access to water, sanitation, and hygiene (WASH) services in unplanned urban settlements and rural fishing villages, continue to derail elimination efforts. Interventions are best planned at a decentralised, community-centric approach to prevent elimination and reintroduction at the district level. Pre-emptive vaccination campaigns before the rainy season and climate-resilient WASH infrastructure in cholera hotspots are also recommended.

**Data availability statement:** All Data is available in the original manuscript submitted.

**Funding:** The author(s) received no specific funding for this work.

**Competing interests:** The authors have declared that no competing interests exist.

## Conclusions/significance

The goal to eliminate cholera by 2025 was unrealistic, as evidence points to the disease becoming endemic. Our findings confirm the need to align health and WASH investments with the Global Roadmap to Cholera Elimination by 2030 through a climate-focused lens. Recommendations for cholera elimination, including improved access to safe drinking water and sanitation, remain elusive in many low-income settings like Zambia. Patient-level information on survival and transmissibility is lacking. New research tailored to country-level solutions and enhancing community participation is urgently required. Insights from this review will be integrated into the next iteration of the National Cholera Control Plan and could apply to other countries with similar settings.

## Author summary

Cholera outbreaks are increasing in both frequency and severity across sub-Saharan Africa, despite long-standing evidence on the effectiveness of improved water, sanitation, and hygiene (WASH), the protective role of oral cholera vaccines (OCV), and the guidance of the Global Task Force on Cholera Control (GTFCC) Roadmap. In Zambia, cholera has become endemic in many settings, yet the true burden remains underreported due to incomplete and inconsistent data. This scoping review synthesises available evidence on the cholera situation in Zambia and identifies critical gaps. It highlights the strong influence of climatic variability, unplanned urbanisation, and fragile WASH infrastructure in driving recurrent outbreaks. Despite the national goal to eliminate cholera by 2025, the findings suggest this target is unrealistic without urgent course correction. The review supports shifting toward a decentralised, community-centric approach to cholera control—emphasising pre-emptive vaccination campaigns, locally tailored WASH investments, and improved surveillance. It also underlines the need for more patient-level research, including on host and environmental factors that influence survival or asymptomatic infection. Findings from this work will inform Zambia's next National Cholera Control Plan and may guide similar efforts in other countries aiming to control or eliminate cholera amid climate and demographic pressures.

## Introduction

Cholera outbreaks are increasing in frequency and severity across the world, particularly in sub-Saharan Africa. This is despite efforts by the Global Task Force on Cholera Control (GTFCC) to achieve cholera elimination in at least 20 countries by 2030 [1]. In 2024, a cumulative total of 804,721 cases and 5,805 deaths were reported across all five regions of the World Health Organization (WHO) in 33 countries [2]. Zambia, with its

Republican President serving as the Global Champion for Cholera Control, had set out to lead the elimination efforts by 2025, ahead of the global targets, with the launch of the first Multisectoral Cholera Elimination Plan (MCEP) in 2018 [3] and a successful pre-emptive Oral Cholera Vaccination (OCV) campaign in 2021 for over five million people living in hotspot areas [4].

However, the country instead experienced its most significant outbreak to date with 23,381 cumulative cases, and 740 fatalities, of which 304 were facility deaths, representing a case fatality of 1.8% (Accessed on 31st July 2024 [5]. A multisectoral response was mounted, including the provision of safe water via water trucking to the hardest hit areas, household chlorine distribution, health education packages, and a reactive oral cholera vaccination campaign [6]. We reported elsewhere a survival analysis of a cohort of patients admitted to treatment centres in Lusaka and found that lack of prior vaccination and the presence of comorbidities were statistically significant contributors to inpatient mortality [7].

The GTFCC Roadmap to Cholera Elimination by 2030 focuses on investment in Water Sanitation and Hygiene (WASH), early case investigation, and the systematic use of OCV as part of cholera elimination strategies as a bridge towards longer-term investments in WASH, health care system strengthening and robust community engagement [1]. The country was conducting a mid-term revision of the MCEP, rendering it necessary to undertake this work to understand what constitutes published knowledge on cholera in Zambia and to learn from lessons and evidence-based practices that could contribute to reduced cholera mortality and the overall number of cases in outbreaks by 2030.

Several other countries earmarked for cholera elimination have documented progress and lessons learned. Haiti, for example, notes the need for Case-Area Targeted Interventions (CATI), given ongoing vulnerabilities and vaccine shortages [8]. In the Democratic Republic of Congo (DRC), a narrative review detailed the successes and challenges in the implementation of three iterations of their National Cholera Control Plan (NCP) (2008–2012, 2013–2017 and 2018–2021) to influence the implementation of their NCP 2023–2027 [9]. They noted that there has been little to no change since the pre-NCP period. Lastly, Uganda noted the use of a scorecard to track cholera elimination efforts at district and ward levels [10]. They highlighted the risks of periods of elimination and then resurgence in some areas if ongoing elimination efforts, such as improved WASH and OCV campaigns, were not sustained [10]. Global efforts to improve vaccine availability and rapid diagnostic kits must be matched by domestic adaptation of GTFCC guidelines to ensure better response efforts during outbreaks and a speedier transition from control to elimination of cholera in endemic countries.

To better adapt cholera control and elimination strategies in Zambia, this scoping review was undertaken to summarise existing evidence on cholera epidemiology and elimination in Zambia, with particular attention to multisectoral and One-Health approaches, incorporating evidence from the human-environment interface. By examining the perceived regression in cholera elimination efforts, we sought to document the evidence generated from the different pillars to facilitate a comprehensive multisectoral response strategy. We addressed two main questions: (i) What is known about cholera in Zambia? (ii) What are the main suggested mechanisms and strategies to further cholera control efforts in the region?

## Methods

A scoping literature search was conducted to identify relevant studies. Our goal was to map the existing literature, present evidence-based strategies in the different thematic areas of prevention/control, and present hypotheses on the best strategy to accelerate progress towards cholera control and eventual elimination in Zambia. We also sought to identify gaps in the research data that could be important for prioritising intervention areas, such as appropriate community-level interventions and evaluation of long-term WASH infrastructure sustainability in rural settings, as outlined in the GTFCC Cholera Research Roadmap [11]. A scoping review was favoured over a systematic review as the goal was to comprehensively map and summarise the existing literature on a broad topic and to identify emerging themes around evidence-based strategies that could accelerate the progress towards cholera control in Zambia.

In preparation for this scoping review, articles were identified in PubMed and Embase using the search terms 'cholera' and 'Zambia' for articles published between 1st January 2013 and 30th May 2024; filtered to English only. The same search terms were used for Google Scholar 'cholera' AND 'Zambia', limited to January 2013 to June 2024. Reference

lists of selected papers and reviews were also screened for relevant papers, as were local publications and preprints within the period under review. Exclusions were made for all conference abstracts, meeting reports, editorial letters, daily situation reports, systematic review protocols or where Zambia or cholera was not mentioned in the abstract. All identified citations were uploaded into a Mendeley database, and data were extracted using a predesigned form. Key findings and study designs were then collated into thematic areas based on the GTFCC Global Road Map for Cholera Control [1]. The GTFCC describes three axes achievable across six different pillars for a comprehensive multisectoral control plan – effective leadership and coordination, surveillance and laboratory, case management, risk communication and community engagement, OCV and WASH [1]. Selected papers were analysed by the theme, and scrutinised for their aims, study design, population, location, identified risk factors and possible mitigative factors. The search was conducted, and all papers were screened between December 2023 and July 2024. Results were synthesised by theme and recommendations.

## Results

### Study identification and selection

A total of 49 records were identified that investigated cholera and Zambia from January 2013 to June 2024 from PubMed, including one previous article exploring the epidemiology of cholera in Zambia from 2000 to 2010 [12]. An additional 23 unique records from a total of 76 were identified from Embase, and 13 more were found on Google Scholar, of which three were unique. Four additional titles were identified from alternative sources, such as preprints, in local journals or otherwise not listed on PubMed, and were included for analysis [13–16]. Full texts were available for all the studies and reported according to the PRISMA-Scoping Review (Scr) guidelines to ensure a systematic and transparent approach to study selection and data extraction, thereby enhancing the reliability, reproducibility and comprehensiveness of the review process [17]. After excluding 22 articles based on the set criteria, the total number of articles included was 53 (Fig 1). Fig 2 shows how the analysed publications were evaluated considering the different pillars they represent. The completed PRISMA-Scr checklist is included in the S1 Table.

### Cholera epidemiology and burden

Table 1 shows studies exploring the epidemiology of cholera in Zambia. Consistently, the definition of an outbreak, based on the national Integrated Disease Surveillance and Response (IDSR) guidelines, was the confirmation by stool culture of *V. cholerae* in at least one cholera suspect patient with three episodes of acute watery diarrhoea in a 24-hour period in each district [18]. Once an outbreak has been declared, the subsequent cholera suspected patients are included in the line list based on the clinical case definition, with or without culture confirmation [18]. Only three studies depict multiyear surveillance data and are represented in Table 1 [6,12,19]. From 2000 to 2010, 39,285 cases in total over the ten years, with 80% of these cases occurring in Lusaka, the capital [12]. Overlapping slightly in years, Mwaba et al described the spatial distribution of cholera cases from 2008-2017 and again found that cases were primarily from Lusaka [19]. They identified 16 other cholera hotspots and noted that outside of Lusaka, cases were mostly identified in districts bordering Tanzania, Mozambique, Malawi or the DRC – suggesting a linkage to the movement of people to and from neighbouring countries [19]. However, at the peak of the 2024 outbreak, 70 of the 116 districts in the country reported confirmed cholera cases with evidence of local transmission [20]. In describing the 2024 outbreak response in Lusaka – the most affected province, Kateule and colleagues depict the epidemiological trends over time since the first outbreak in 1977 and demonstrate an increasing magnitude of cases but a reduction in the case fatality rate from close to 10% in 1978 to about 3% in 2024 [6]. Before 2013, the largest outbreaks had encompassed 13500 cases in 1991 and 1999. There were no large-scale outbreaks recorded after the launch of the MCEP in 2018 until the 2024 outbreak, which was the largest [6]. A limit may be that cholera cases were only reported in the IDSR system when an outbreak is confirmed by culture.

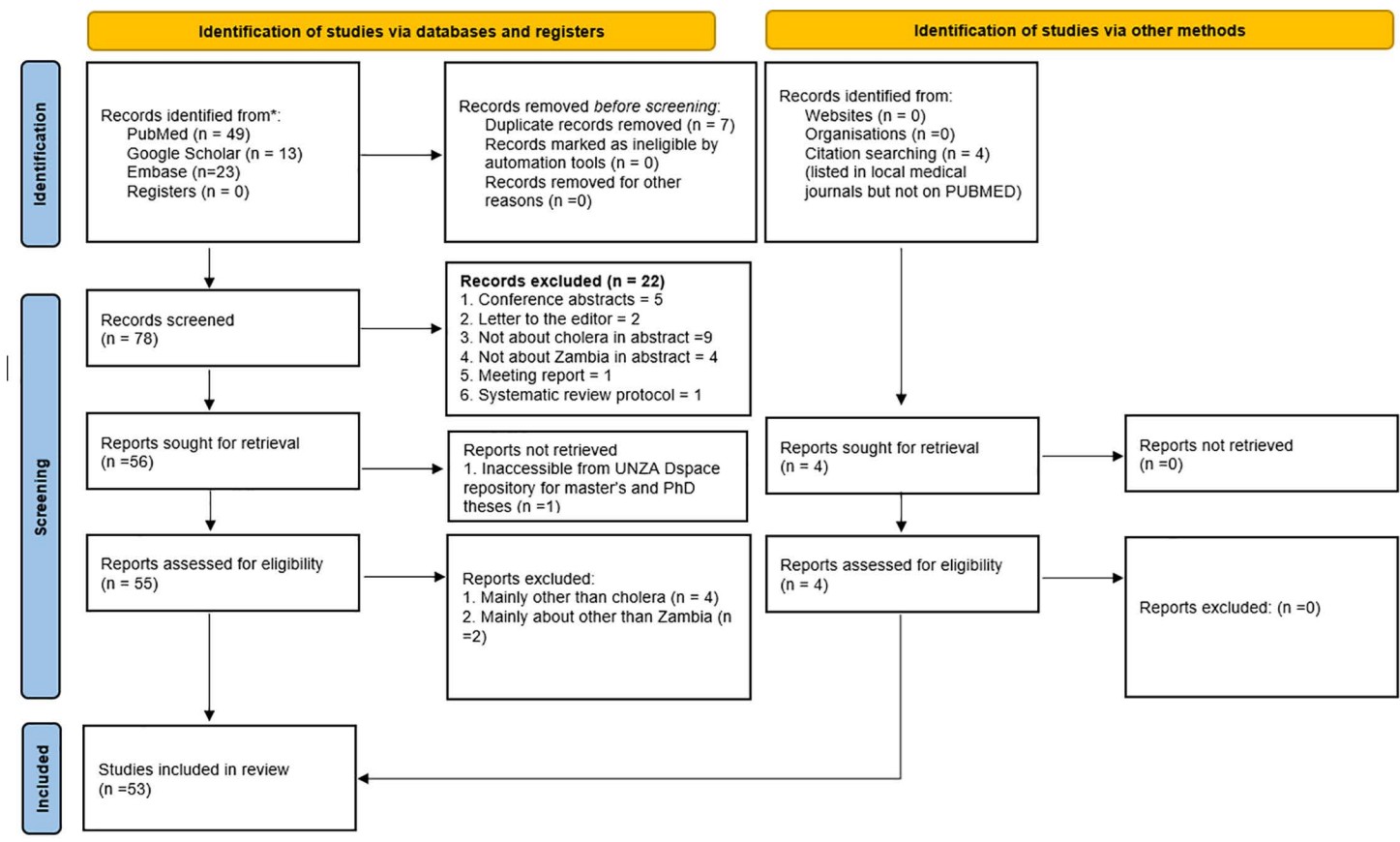

*No automation tools were used, all exclusions were done by human

**Fig 1. Study flow diagram showing the selection processes in line with the PRISMA-Scoping Review (Scr) Guidelines:** Fig 1 shows the flow diagram illustrating the scoping review process according to PRISMA guidelines, highlighting selection and inclusion criteria, search strategy, screening, eligibility assessment, and final included studies.

Consequently, as documented by Wiens and colleagues, in endemic areas where suspected cases are not routinely subjected to laboratory confirmation, the true incidence and overall disease burden may be significantly underestimated, particularly outside of outbreak season [21]. Additional studies attempted to explore risk factors of the outbreaks, the regions and age groups of affected individuals, but were limited in size and scope [7,13,15,22–24]. Fig 2 shows the bias towards descriptive analyses of each localised outbreak.

### Risk factors and determinants of transmission

Male sex, close contact with a cholera case and the use of borehole water were found to be risk factors for cholera infection [14,23–28]. Drinking water sources were found to have inadequately low free-residual chlorine (FRC) in up to 71% of households surveyed [14,25,28]. Thirty-one per cent of those households with inadequate FRC had evidence of faecal contamination. Low latrine coverage, poor drainage systems, and sharing latrines [14,24] were also documented vulnerability factors that allowed for the perennial occurrence of cholera in some localities, particularly unplanned settlements such as the fishing villages in many of the areas bordering lakes and in high-density, peri-urban communities of Lusaka and the Copperbelt [19,22,26–28]. Whilst poor hygiene practices (mostly superimposed on people due

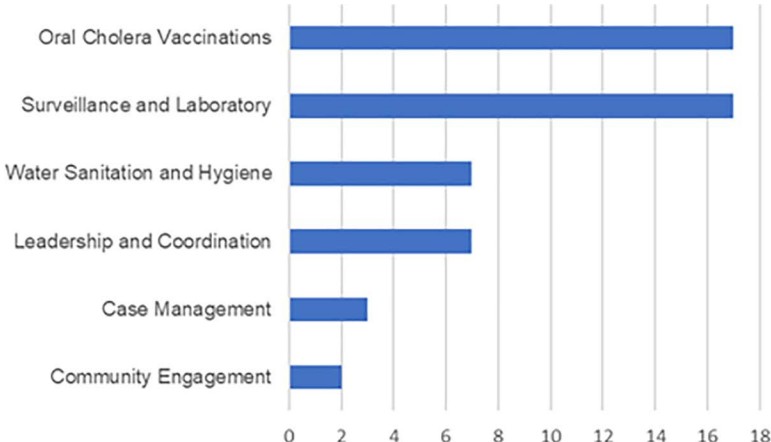

**Fig 2. Frequency of *Cholera* Publications *in Zambia* by theme from 2013–2024** **Fig 2 illustrates the distribution of cholera-related publications in Zambia from 2013 to 2024 categorised by thematic focus.** Themes include epidemiology, public health interventions, water, sanitation, and hygiene (WASH), healthcare infrastructure, and community engagement. The data provides insights into the evolving research priorities and strategies aimed at combating cholera within the Zambian context over the past decade.

**Table 1. Cholera Epidemiology and Burden in Zambia (2013–2024).**

| Author | Aims | Study Design | Number of Participants | Location |
|---|---|---|---|---|
| Olu et al. 2013 [12] | Cholera epidemiology in Zambia from 2000-2010 to describe implications for improving cholera prevention and control strategies in the country | 10-year descriptive data extracted from the electronic IDSR compared to the Global Health Atlas to ensure consistency. Subsequent review of the minutes of the National Epidemic Preparedness and Prevention Committee Meetings and key informant interviews with stakeholders | In 2010, 6794 cases (a 500% increase compared to 2003) and 115 deaths (CFR 1.6%). 39,285 cases in total over the 10 years | Seventy-three per cent (73%) of all the cases of cholera reported from 2001 to 2010 were from Lusaka Province, followed by 7%, 6%, 5% and 4% respectively from Luapula, Southern, Central and Copperbelt Provinces |
| Mwaba et al. 2020 [19] | Identification of cholera hotspots in Zambia: a spatiotemporal analysis of cholera data from 2008 to 2017 | Descriptive analysis using Poisson-based space and time scan statistics to account for spatial differences | 34,950 participants countrywide; however, 29080 of these were in Lusaka over the 10 years | Country-wide, 80% of the cases were Lusaka-based |
| Kateule et al 2024 [6] | Multisectoral approach for the control of cholera outbreak - lessons and challenges from Lusaka district - Zambia, October 2023 - February 2024 | Descriptive observational study of cholera response activities in Lusaka district. Mixed method qualitative and quantitative with review of surveillance records and documented response interventions and challenges using situation reports | Lusaka District with 13122 cases and 498 deaths as of 12th February 2024. They demonstrate the increasing magnitude of cholera outbreaks in Zambia. Prior to 2013, the largest outbreaks had 13500 cases in 1991 and 1999. | Focus on the six subdistricts in Lusaka where the epicentre of the 2024 outbreak occurred. |

to lack of facilities) were a notable risk factor, consumption of food products, particularly fresh fish, was not associated with an increased risk [16].

The availability and quality of drinking water in the peri-urban areas of Lusaka were assessed [28]. It was found that in areas underserved by the municipal utility companies, private borehole companies known as 'Water Trusts' would operate small shops known as 'kiosks' where community members could go and draw small quantities of water in buckets at a minimal cost to cover the fees only, and not for profit [28]. These trusts treated and provided water to the communities in these water-stressed unplanned settlements as an adjunct to the provincial utility company, and yet they were found to serve less than 60% of the communities in need of their services [12,24]. Despite this limitation, they were noted to

present a safer alternative than privately owned consumer boreholes and shallow wells in terms of faecal contamination with *Escherichia coli* and nitrite content of the water [28]. Those unable to afford the kiosk water tended to use unsafe surface water sources such as shallow wells in their locality [24,25]. These presented the highest risk of contamination, particularly due to topographical features such as the high-water table in Lusaka Province, leading to a high risk of contamination of these shallow wells from nearby pit latrines [27,28].

The smallest surveillance unit of population reported was the ward level. It was found that the greatest risk for cholera was in the wards with the densest populations, unimproved sanitation and evidence of *E. coli* contamination of piped sources [14]. Elsewhere, as was seen in Kabwe, through an environmental sampling of groundwater using polymerase chain reactions (PCR) tracers, there was evidence of groundwater contamination with environmental vibrio [27]. The authors postulated that private boreholes are vulnerable to contamination, possibly due to incompetent casing, which may provide an artificial pathway for the Vibrio from contaminated ground sources and pose an even greater risk. Supporting this was the rapid decrease in cases seen during outbreaks, when there was an increased provision in WASH services, such as hyper-chlorination of the water utility lines, provision of safe water through emergency tanks in the hotspot areas [29] and with the use of reactive OCV campaigns [30].

### Inter-district and inter-country spread of outbreaks

Risk factors for continued outbreaks between the peak years included increased poverty and inadequacies of social services due to rural-urban migration [12,24,31]. Similarly, movement between neighbouring districts [15,29] and neighbouring countries [32–35] was identified as a factor associated with epidemic cholera in Zambia. Chirabombo and colleagues documented how naïve districts neighbouring traditional hotspots such as Lusaka can present with outbreaks of their own, with evidence of local transmission [15]. The question of environmental persistence versus reintroduction into the district from neighbouring countries such as the DRC and Tanzania, which equally have continuous outbreaks, has been documented [33–37]. This is reaffirmed by laboratory studies and descriptive analyses of genomic sequencing isolates that showed a wide genetic diversity [33–34] and close linkage with isolates from other parts of the Great Lakes region [35–37]. This underscores the need for both a decentralised approach at the district and ward levels, but also shows a need for enhanced cross-border surveillance and possible cross-border joint responses [25–38].

### Clinical characteristics and host predisposition

Globally, there is a dearth of information on the clinical characteristics of patients affected by outbreaks beyond general case counts and case fatality rates [39]. Little is known about the proportion of pregnant women, elderly or paediatric patients affected by cholera, nor the number of patients presenting with co-morbid conditions or other complications of care. What was seen is that having received limited education and being older than 55 years constituted one risk factor for increased mortality [23]. There was a slightly higher proportion of patients documented to have died before arrival at the treatment facilities (i.e., at home or community deaths) versus in the facility in the 2024 outbreak in Lusaka (60% community deaths) [6], proportionately more than the community deaths reported in the 2018 outbreak (45%) [23]. Intravenous fluids were not available beyond Cholera Treatment Centres (CTC), and there was no documented use of Community Oral Rehydration Points before the 2023/2024 outbreak [6]. Adequate Oral Rehydration Solutions (ORS) was protective [23], yet it was clear that there were disparities in the availability of ORS, particularly in rural communities [40]. Rising antimicrobial resistance was found to have direct implications on patient management in cholera treatment settings [1]. Specifically, it can compromise the effectiveness of antibiotics used for managing severe cases, potentially leading to prolonged illness, increased risk of complications and greater strain on clinical resources [38]. Prior antibiotic use was not found to be protective, although noted that patients often took metronidazole, which is not one of the recommended agents [23]. In younger patients, cholera was noted to be an important cause of morbidity and mortality in the under-five age group, with increasing antimicrobial resistance over the years [32,41,42]. For example, earlier studies showed

low-level resistance to tetracyclines but as high as 95% in subsequent outbreaks due to its use as drug of choice for first-line treatment of severe patients [38,42]. Case management was reported to have improved, with reductions in the case fatality rate (CFR) decreasing from 6.7% in 2000 to 1.7% in 2010 [12]. However, as of the 2018 outbreak, the case fatality rate hovered around 2.5% [25]. The case fatality rate of the 2023/2024 outbreak was 1.3%, with increased documentation of community deaths [3]. For inpatient fatalities, there were higher odds of dying for those with pre-existing comorbid conditions [7].

## Vaccine availability and effectiveness

Vaccines are known to be a useful tool for community-level interventions for controlling waterborne diseases such as cholera, in places where access to water, sanitation and hygiene remains limited [43,44]. Recent studies have used the new Euvichol Plus, which is the Eubiologics bivalent vaccine of El Tor and Ogawa, presented in glass containers as opposed to plastic vials to improve cold chain in humanitarian crises [45]. They have shown a higher vaccine efficacy in the two-dose strategy than the single dose (at 74% and 81%, respectively [46,47]; and that reported OCV administrative coverage is often much higher than the actual coverage which was found to be 66% of people getting both doses, which may further lower efficacy rates [48]. Questions persist about the very high dropout rate of 18% between the two doses [48]. Similarly, it remains to be seen the effect of previous preventative campaigns, as a lead-up to future multi-year preventative vaccination campaigns, or if delaying the second dose post outbreak can be used to time subsequent campaigns before the rainy season, which is a high-risk period for cholera transmission [49].

Pugliese-Garcia and colleagues attempted to explore the factors influencing vaccine acceptance and hesitancy in the hotspot districts of Lusaka. They found that traditional remedies, religious beliefs and alcohol use persist as impediments [50], as does a background mistrust towards Western medicine [51]. There was an overarching sense of helplessness or 'fate' as the participants were aware they could not change their living conditions and did not realise their ability to use safer water practices to protect themselves [51].

Investigation of the immunogenicity of the vaccines in a controlled population in one of the high-risk fishing villages found no significant difference in vibriocidal antibodies at two weeks or six months and provided evidence for the delayed dosing schedule [52] but also waning immunity beyond 12 months [53]. The group found no influence of ABO blood groupings on vaccine response [54]. HIV-positivity was found to reduce immunogenicity in these individuals regardless of the CD4 count, whilst serum vitamin A levels had no effect positive or negative [55]. Elsewhere, there was a suggestion of vitamin A supplementation as a possible adjuvant to improve T-cell expression following vaccination, particularly in children [56], which may offer a gateway into host-specific factors for improved immunity and transmission dynamics. There was no work yet published on the role of the gut microbiome in cholera vaccine responsiveness or protection in the face of household exposure. However, a review article describing environmental enteric dysfunction (EDD), a subclinical disorder of intestinal function in settings of poverty that affects vaccine uptake, concluded that the immunogenicity and efficacy of oral vaccines in developing countries was less in developing countries than developed countries based on pathology findings [57] The evidence surrounding EDD in Zambian cohorts is limited but also points towards a potential role of Immunoglobulin A supplementation to improve uptake of vaccines through improved nutritional status [58]. Most recently, a comparison of vibriocidal antibodies in naturally infected vs vaccinated individuals was found to be comparable, with peak immunity seen around day 19 post-infection and waning after day 30–39 [42] . The group explored waning immunity beyond 90 days in revaccinated individuals compared to naïve and found that repeated use of a single dose strategy was unprotective and probably contributing to more explosive future outbreaks following such campaigns [59]. suggested the need for booster vaccinations, particularly in high-risk areas, as a possible public health protective strategy. Additionally, ongoing work is being done to explore Zambian Vibrio cholerae strains for human challenge studies, to explore future vaccine candidate efficacy [60].

The use of a single-dose campaign of Sanchol was found to be cost-effective, amounting to just under $ 1 million to vaccinate 500,000 people [30]. A further evaluation of the cost of cholera illness and the cost-effectiveness of the single-dose campaign in Lusaka was close to $1000 per disability-adjusted-life year (DALY) averted, especially in those above the age of 15 years [61]. The social implications for affected communities have not been deeply studied, nor the cost-benefit analysis of community-based interventions and health education initiatives in the hotspot districts. With the increasing size of the outbreaks, it remains to be seen the cost-effectiveness of reactive campaigns, and also the macro-economic effects of the overall cholera responses.

## Climate variability

The role of climate variability and extreme weather events cannot be ignored, with a strong association between the onset of rainfall and epidemic outbreaks [25]. Cholera outbreaks in Zambia, like many other African countries, are seasonal [25,36], differing from the Ganges Delta, where it occurs perennially [36]. The outbreaks start with the onset of the rainy season in 71% of cases and have been associated with 50% of all recorded drought years. Outbreaks are expected to increase in frequency by 300% in the near future with recurrent El Niño events [62]. El Nino events are associated with increased rainfall and flooding, which would lead to contamination of water sources, whilst the warmer temperatures will allow the growth and persistence of Vibrio cholerae in the environment. These conditions together create a conducive setting for the spread of cholera in vulnerable communities [62]. Following seasonal rains, the larger outbreaks are often heralded by flooding, which is a specific sequel of torrential rains possibly enhanced by climate change. Flooding has been associated with damage to WASH infrastructure, and the decay of flooding countermeasures, such as clogged-up drainage canals and sealing of ground passages for water, particularly when big cities such as Lusaka are afflicted [25,62], further compounding the problem. Reduced rainfall (i.e., drought periods) may also increase cholera outbreaks as seen in the U-shaped occurrence of diarrhoeagenic bacteria such as *V. cholerae* with rainfall and pathogen proliferation, meaning an increase in both ends of the spectrum – very dry and then very flooded – can contribute to increased cholera incidence [63]. Groundwater drilling during the drought years, if not carefully planned, will worsen the already water-stressed situation in certain parts of the country [62]. The anticipated periods of droughts in the near future are expected to exacerbate rural-urban migration into the peri-urban slums, further compounding the water-stressed situations and the likelihood of larger cholera outbreaks [62].

Mathematical modelling was used to predict the expected time to extinction of cholera in Lusaka, and based on previous estimates of a second wave in each outbreak found that heavy rains were associated with an increased environment-to-human transmission [31]. They warned that environmental vibrio could persist for eight months to six years in the environment, especially the shallow wells and areas with poor drainage, hence future outbreaks would be longer and more severe. They also recommended enforcement of the multisectoral cholera elimination plan, which sought the combination of WASH interventions with periodic oral cholera vaccinations [31]. A study exploring microbiological screening of plankton and meteorological monitoring of Uvira in DRC and Mpulungu in Zambia between 2000–2014 to better understand environmental factors that trigger cholera outbreaks in the region, concluded that whilst climate dynamics play a part in cholera transmission, most outbreaks in Africa region are due to genetically diverse strains that spread into non endemic areas and cause explosive outbreak [64] They suggested the need for localised prevention efforts to protect communities from introduction of new outbreaks, a nod to decentralisation of cholera control and prevention efforts. Chota and colleagues attempted to draw lessons from the cholera outbreak of 2017–2018 when responding to the COVID-19 pandemic in 2020–21. They engaged health care professionals and community leaders in focus group discussions around the successes and pitfalls of multisectoral response strategies. They concluded that challenges in the partnership collaboration included inadequate resources, poor communication, poor coordination, lack of clear shared vision, reactive response, poor involvement of the community, hegemonic powers and mistrust of each other [65]: "Despite the attempts at co-ordination, ministries have a tendency of operating in isolation, this has resulted in lack of a clear shared

vision. This also contributes to duplications of tasks in trying to prevent an outbreak of cholera" [65]. It has been proposed that the key to success in cholera elimination would be greater community participation in developmental activities and empowering the communities to take ownership of their health by addressing underlying economic challenges [66]. All the reviewed articles are listed in Table 2 with their key findings and possible mitigating factors that can contribute to cholera control and elimination in Zambia.

## Discussion

Since the first documented outbreak in 1977, Zambia has recorded major outbreaks every three to five years with increasing intensity and fatality [6,12,19]. The outbreaks were predictable concerning the timing in the calendar year and with an increasing frequency related to climatic conditions and urbanisation [14,23,27]. Because most of the reporting is done based on case definitions during outbreaks, it is postulated that the true burden of cholera in Zambia, like other parts of the world, is underreported outside of explosive outbreaks [21,67]. The major risk factors for recurrent outbreaks in the country were poor access to water and sanitation services in urban unplanned settlements and the rural fishing villages [19,24,25]. These factors were found to be persistent even in the 2023/2024 outbreak, which is the largest to date [6], different from other cholera-prone areas, which are often coastal areas in South Asia [36] or places with humanitarian crises and conflicts, such as Northern Nigeria and Haiti [68,69].

Zambia was not considered endemic to cholera at the time of the development of the first Multisectoral Cholera Elimination Plan (MCEP) in 2018. However, the increased frequency and near-annual occurrence of outbreaks in certain localities now justifies its reclassification as a cholera-endemic country, eligible for sustained cholera control rather than elimination, in line with GTFCC guidance [1]. Given the documented risk of cholera re-introduction across wards and districts due to population movement, as demonstrated in Lusaka, where transmission events occurred across multiple peri-urban areas [32,33], interventions should prioritise a decentralised, community-centric approach to surveillance, case management and community engagement [65]. Case-Area Targeted Interventions (CATI), which support rapid, localised response to confirmed cases, have demonstrated operational effectiveness in comparable high-risk settings such as Uganda, the DRC, and Burundi, and are increasingly recognised as effective components of cholera elimination strategies [9,70,71]. The predictable geographic location and seasonality of the outbreaks could be used to envisage the location and size of repeat vaccination campaigns, with the possibility of pre-emptive campaigns timed before the rainy season to be included in the expanded program for immunisations, as has been demonstrated in Indian cohorts [72–74]. Excitement surrounds the recent WHO prequalification of Euvichol-S, a simplified version of the Euvichol-Plus that is easier to produce but equally efficacious [75]. It is anticipated that its inclusion in the global stockpile will increase vaccine availability, enabling countries like Zambia to implement multi-year vaccination campaigns as part of the cholera control and elimination efforts. These multi-year vaccine campaigns would serve as a bridge to increased WASH investments. Similarly, WASH infrastructure should be planned in a decentralised framework construct as the different localities, even within a single country, face unique vulnerabilities, which are expected to intensify with evolving climate patterns [10,19,24,62,63,76]. Ultimately, cholera elimination would depend on approaching prevention from a developmental lens and not outbreak response. This entails building resilient communities with available community resources, effective communication, local knowledge, training and education [76,77].

Efforts to combat vaccine hesitancy must be sustained and embedded within long-term public health strategies, rather than implemented reactively during outbreaks [65]. Persistent myths and misconceptions, often stemming from historical injustices, socio-political marginalisation or fears of Western exploitation and medical malevolence, require culturally sensitive and community-led approaches to effectively address [50,51,77]. This is particularly relevant in contexts where mistrust in health systems continues to shape public perceptions of vaccination campaigns [50,51,65]. Building public trust demands continuous engagement through transparent communication, collaboration with local leaders, and integration of behavioural and social sciences into health programming [65,66]. Improving community literacy levels in African

**Table 2. Comprehensive Review of Cholera Research in Zambia (2013-2024): Aims, study design, population, identified risk factors and mitigative measures.**

| First Author | Aims | Study Design | Population and Location | Identified Risk Factor | Possible Mitigative Factor | Theme |
|---|---|---|---|---|---|---|
| Mshana SE 2013 [38] | Antimicrobial resistance in human and animal pathogens in Zambia, DRC, Mozambique and Tanzania | Review of published and unpublished laboratory data from four SSA countries reviewing 68 articles between 1990 and 2020 | Zambia, DRC, Tanzania, Mozambique | Diarrhoea diseases cause 25% of Under5 mortality, with increasing resistance between outbreaks of cholera. for example, the cholera outbreak in 1990 was susceptible to tetracyclines with only 5% resistance, compared to 95% in1992 | Urgent need for sustainable surveillance system and cross-border collaborations. Concluded the need for improved surveillance between countries and antimicrobial resistance and sensitivity data sharing. | Laboratory/ Surveillance |
| Olu O 2013 [12] | Cholera Epidemiology in Zambia from 2000-2010: Implications for improving cholera prevention and control strategies in the country | Review of epi data disaggregated by province from 2000-2010 | Endemic cholera in Lusaka, Luapula, Southern and Copperbelt. In 2010 increased to 6974 cases which was a 500% increase from 2003. CFR 1.6% | Cholera was endemic even at that point with confirmed outbreaks every year. LSWC had set up water kiosks to improve access at $0.02 per 20l of water, which had not increased yet the poor in the slums are still unable to afford the water and resort to shallow wells | Health systems strengthening, multisectoral collaboration and attention to urban development vis a vis well-planned urban dwelling | Leadership & Coordination |
| Moore S 2015 [33] | Relationship between distinct African cholera epidemics revealed via MLVA haplotyping of 337 vibrio cholera isolates | Microbiological screening of plankton and meteorological monitoring to help better understand environmental factors that trigger cholera outbreaks in the region | Uvira in DRC and Mpulungu in Zambia between 2000–2014 47 environmental samples from Mpulungu from water, plankton and fish between August 2012 and October 2014 | Found distinct MVLA haplotypes in the outbreaks different from the singletons found in the environmental samples. Concluded that humans remain the main reservoir disputing the environmental perseverance | Climate dynamics play a part in cholera transmission but most outbreaks in African regions are due to genetically diverse strains that spread into non-endemic areas and cause explosive outbreak | Laboratory/ Surveillance |
| Sorensen JP 2015 [27] | Tracing enteric pathogen contamination in sub-Saharan Africa groundwater | Examined water samples from 22 groundwater supplies in Kabwe and explored 16S RNA gene fragments | In Kabwe boreholes and shallow wells using qPCR as tracers for groundwater contamination | Found new evidence that boreholes are vulnerable to contamination possibly due to incompetent casing of shallow wells providing artificial pathways. 41% of the water sources tested had VC on PCR. First evidence of inland persistence of VC even during off-season | Expansion of the city with informal settings and increased boreholes puts people at an increased risk of contaminated water due to the false assumption that the borehole water is safe First evidence for perennial inland freshwater reservoir of vibrio cholera inland (most were in the tropics – unlike the Ganges delta) | Water, Sanitation and Hygiene |
| Plisnier PD 2015 [64] | Cholera Outbreaks at Lake Tangayinka induced by Climate change.WASH | Microbiological screening of plankton and meteorological monitoring of Uvira in DRC and Mpulungu in Zambia between 2000–2014 to help better understand environmental factors that trigger cholera outbreaks in the region | 47 environmental samples from Mpulungu from water, plankton and fish between August 2012 and October 2014 | Found distinct MVLA haplotypes in the outbreaks different from the singletons found in the environmental samples. Concluded that humans remain the main reservoir | Climate dynamics play a part in cholera transmission, but most outbreaks in African region are due to genetically diverse strains that spread into non endemic areas and cause explosive outbreaks | WASH |

*(Continued)*

**Table 2.** (Continued)

| First Author | Aims | Study Design | Population and Location | Identified Risk Factor | Possible Mitigative Factor | Theme |
|---|---|---|---|---|---|---|
| Matapo B 2016 [13] | Successful Multi-parter response to a cholera outbreak in Lusaka, Zambia 2016: a case-control study | Case-control study done 1:3 to identify factors associated with a cholera outbreak | Cases were identified from the CTC register in Bauleni and controls were residents of Bauleni without water diarrhoea between March and May 2016 | Positive vibrio in stool was associated with drinking inadequately treated bore-hole water. | Akin to the London cholera outbreak of 1854, the closure of a contaminated water source (borehole in this case) directly led to a reduction of cholera cases in the locality. Vaccines were also deployed | Leadership & Coordination |
| Mwaba J 2016 [41] | Evaluation of the SB Bioline cholera rapid diagnostic test during the 2016 cholera outbreak in Lusaka, Zambia | RCT of RDT vs Culture on fresh stool of 170 cholera suspects. | Lusaka-based RCT during the 2016/2017 outbreak between April and June 2017. 90% sensitivity and 95% sensitivity and recommended for use in the field early in an outbreak | Paper focused on the lab analysis of the RDT kit and found it as a useful tool to increase the turn-around time for surveillance needs especially in settings with prevalent acute water diarrhoea | Enhanced surveillance for cases with AWD to increase the identification of cases | Laboratory/Surveillance |
| Chirambo RM 2016 [15] | Epidemiology of the 2016 cholera out-break in Chibombo District, Central | Descriptive study using routine epi-data | 23 cholera cases meeting case definition between 9th Feb - 20 March 2016 | Index case was imported from Lusaka hence even areas that have never reported outbreaks can have introduction | More emphasis needed on preventative efforts as opposed to response efforts. | Laboratory/Surveillance |
| Mwambi P 2016 [26] | Timely response and containment of 2016 Cholera out-break in Northern Zambia | Descriptive study of epidemiological records | 68 cases from Nsumbu, Nsama district between 10th March and 3rd April 2016 | Outbreak was precipitated by flooding of Kapisha Dam, leading to the submerging of pit latrines. High CFR of 3.6% possibly due to late notification | Improve response timeliness (utility of 7-1-7 frameworks) | Laboratory/Surveillance |
| Chiyangi H 2017 [42] | Identification and antimicrobial resistance patterns of bacterial enteropathogens a prospective cross-sectional study | Hospital-based cross-sectional study examining stool samples | Cross-sectional study of stool in children 0–59months, enrolled 271, December 2015 to April 2016 at University Teaching Hospital in Lusaka | 31% of total samples had either VC, Salmonella, DEC or Shigella. (40.8% of which were VC). Of the cholera cotrimoxazole resistance and the common pattern | Was part of the recommendation to change our case management guidelines for Cholera Case Management guidelines away from empirical cotrimoxazole and ampicillin to Doxycycline | Laboratory/Surveillance |
| Gama A 2017 [22] | Cholera Outbreak in Chiengi and Nchelenge Fishing Camps, Zambia 2017 | Outbreak report reviewing medical records of suspected and confirmed cases | 76 cases from Nchelenge and Chiengi districts in Luapula Province which neighbours DRC on the shores of Lake Mweru | Poor sanitary facilities in the fishing camps, movement of people across water borders, poor knowledge of WASH and drinking water from shallow water sources that were contaminated | Recommended improved lab capacity even in rural settings to improve case confirmation and surveillance. Continuous community sensitization | Laboratory/Surveillance |
| Ferreras E 2018 [44] | Single dose cholera vaccine in response to an outbreak in Zambia, correspondence in the NEJM | Matched case-control study to quantify the short-term effectiveness of a single dose campaign. Between April 2016 and June 2016 | 66 cases with confirmed cholera and 330 matched controls from Lusaka | This was the early evidence of the protective efficacy of short-term reactive campaigns with OCV. Still, the fact that we had another outbreak in Lusaka in 2017 makes one question the efficacy of the single-dose campaign | Probably OCV isn't the only answer to achieve cholera elimination. especially since shortages on the global stockpile don't seem to be ending soon. Although suggested doing so annually before the endemic season, like in Cameroon | Oral Cholera Vaccinations |

*(Continued)*

**Table 2.** (Continued)

| First Author | Aims | Study Design | Population and Location | Identified Risk Factor | Possible Mitigative Factor | Theme |
|---|---|---|---|---|---|---|
| Kapata N 2018 [29] | A multisectoral emergency response approach to a cholera outbreak in Zambia from October 2017 to February 2018 | Descriptive analysis of outbreak response with timing of particular interventions | Cholera outbreak in Lusaka, between 7th October 2017 and February 2018 when 3989 cases reported | Poor drainage, reliance on groundwater, and showed how the deployment of emergency tanks reduced the number of cases. Reported also on the first use of OCV in a reactive campaign to help curtail the outbreak | advocated for the first iteration of the MCEP with a combination of planned pre-emptive campaigns of OCV every 3 years, and longer-term WASH investment | Leadership & Coordination |
| Mwanza Lisulo M 2018 [56] | Retinoic acid elicits a coordinated expression of gut homing markers on T lymphocytes of Zambian men receiving oral Vivotif but not Rotarix, Dukoral or Opvero vaccines | Initial paper was mouse models for supplementation of vit A and immunogenicity (52), second phase was in adult males | Diminished immunogenicity and efficacy of oral vaccines, Vit A supplementation reduces death | The immune boosting effect of vitamin A is in typhoid vaccine but not the others | Immune boosting effect of vit A seen in typhoid vaccine but not others including Duchoral and Rotarix however this shows insight into the possibility of adjuvants to improve host response to the vaccines | Oral Cholera Vaccinations |
| Poncin M 2018 [30] | Implementation research: reactive mass vaccination with a single dose oral cholera vaccine, Zambia | Documented the successful implementation of a single dose campaign in 10 wards of Lusaka. Single dose because of shortage on the global stockpile | Lusaka during the 2016 outbreak, with a campaign conducted 2 months from confirmation of the first case | Showed successful implementation of the single dose campaign (though it cost almost $1mil total for the campaign to vaccinate just over 424,100 doses given in 17 days representing 78% coverage) meaning cost increases with every outbreak - currently on 3 mil doses needed for Lusaka. Plus an outbreak 2 years later questions the efficacy | Increase access to vaccines to keep the global stockpile replenished. In the meantime, single-dose campaigns allow protection for a greater population during an outbreak. Also, static sites more effective than house-to-house | Oral Cholera Vaccinations |
| Pugliese-Garcia 2018 [50] | Factors influencing vaccine acceptance and hesitance in three informal settlements in Lusaka, Zambia | Nested in vaccine uptake study | Reported findings from 48 focus group discussions with lay ppl, neighbourhood health committee members and vaccinators | Traditional remedies, alcohol use and religious beliefs emerged as drivers of vaccine hesitancy, likely reinforced by a background of distrust towards Western medicine | recommended community-driven models that incorporate factual communication by professionals. Vaccine information should be pre-emptive not just during the campaigns | Community Engagement |
| Church AJ 2018 [58] | Exploring the Relationship Between Environmental Enteric Dysfunction and Oral Vaccine Responses | Review article | 8 papers identified exploring EDD and vaccine efficacy | there was substantial heterogeneity in study design and few consistent trends emerged. Four studies reported a negative association between EED and oral vaccine responses; two showed no significant association; and two described a positive correlation. | In Zambian cohorts described in the study they suggested IGA supplementation (particularly for rotavirus) but explained that data was limited and more work would need to be done | OCV |

*(Continued)*

| First Author | Aims | Study Design | Population and Location | Identified Risk Factor | Possible Mitigative Factor | Theme |
|---|---|---|---|---|---|---|
| Marie C 2018 [42] | Pathophysiology of environmental enteric dysfunction and its impact on oral vaccine efficacy | Review article describing environmental enteric dysfunction, a subclinical disorder of intestinal function in settings of poverty that affects vaccine uptake | Review | Immunogenicity and efficacy of oral vaccines in developing countries less in developing countries than developed. Example given was Vacchora. Explained also that EDD is common in our setting and gave examples of pathology results from Zambia | While there is important evidence from ecological studies that EED and oral vaccine failure are associated, rigorous proof in multiple populations is lacking. If effective therapy were available for any of the domains of pathophysiology of EED, it would be possible to demonstrate that such therapy improves responses to oral vaccines. Such therapy is not yet available, but it is likely that it would also improve child growth and possibly micronutrient status. | OCV |
| Sinyange N. 2018 [25] | Cholera epidemic-Lusaka, Zambia October 2017 - May 2018 | Description of the outbreak and the interventions done at different stages of the response. A cross-sectional household survey | 5,905 cases and 98 deaths case fatality rate of 1.6% Knowledge, Attitudes and Perspectives (KAP) survey in 98 households in the affected communities | Inadequate supply of safe water by utility companies, i.e., use of shallow wells, private boreholes or water kiosks, contamination of piped water sources | Due to cost implications of city-wide water and sanitation infrastructure, a targeted approach to improvement in the particular wards with firstly flush to sewage sanitation systems, and then possibly improvement of piped water sources in these communities. | Laboratory/Surveillance |
| Ferreras E 2019 [49] | Delayed second dose of OCV administered before high-risk period for cholera transmission: cholera control strategy in Lusaka, 2016 | Post vaccination coverage survey done in December 2016 following the outbreak that was declared in February (Poncin 2018 reported on the actual campaign) | 505 randomly selected people after 1st round, and 442 after 2nd round | Post-vaccination coverage survey only 33.9% of two doses, and 36.0% of one dose in the targeted neighbourhoods. And for those getting their vaccination in April 30% vs 70% in December suggesting that only a fraction of the population was still present in the vaccination areas. Only 19% had two doses and there was another outbreak in October 2017 showing limited efficacy of the targeted campaign | They had suggested that annual campaigns prior to the cholera seasons might be a more effective strategy to reduce the risk of outbreaks in places at high risk of transmission, especially in settings like this with highly mobile populations | Oral Cholera Vaccinations |
| Heyerdahl LW 2019 [51] | "It depends on how one understands it" - a qualitative study on the differential update of OCV in 3 compounds in Lusaka | Study on community perspectives of OCV, nested study within the rapid qualitative assessments in 3 compounds in Lusaka during the 2016 outbreak | Findings from 18 focus group discussions with equal men and women who reported being unvaccinated during the first and second round of vaccinations and 6 with men and women who were vaccinated at the end of the second round | Some in at-risk groups not taking the vaccine due to concerns of Western malevolence. Those who took both doses had awareness of their risks and that they were unable to change their living conditions. Others though did not take the vaccine because they felt helpless and susceptible anyway | Myths and misconceptions exist that could affect vaccine acceptance, some steeped deep in traditional beliefs, so need to be more transparent and open communication, and more local studies on efficacy | Community Engagement |

*(Continued)*

PLOS Neglected Tropical Diseases

**Table 2.** (Continued)

| First Author | Aims | Study Design | Population and Location | Identified Risk Factor | Possible Mitigative Factor | Theme |
|---|---|---|---|---|---|---|
| Tembo T 2019 [61] | Evaluating the cost of cholera illness and cost-effectiveness of a single dose OCV in Lusaka Zambia | Retrospective cost-effective analysis to estimate out-of-pocket costs to the individuals who were treated for cholera | From April to June 2017, 189 cholera survivors from Lusaka | The cost per administered vaccine was US$1.72, treatment costs higher for older patients $17.66-$35.16 mostly for non-medical items. Costs per case averted by vaccination $369-$532, cost per life year saved US$18515 - US$27,976 and total DALY averted was up to $1000 for patients older than 16 | Cost-effectiveness of the reactive vaccination campaign, particularly at the household level but not on a macroeconomic scale | Leadership & Coordination |
| Ferreras E 2020 [46] | Alternative observational designs to estimate the effectiveness of one dose OCV in Lusaka, Zambia | Compared the methods of testing effectiveness, matched case-control, test negative case-control and case-cohort study to interrogate methods of vaccine efficacy studies | 360 vaccinated and 561 unvaccinated individuals in Lusaka. Followed up for 6 months | found 88% effectiveness of one dose strategy but for only 60 days protection. Useful in reactive campaigns. Bias towards elderly patients. Poor quality of vaccination cards so poor retention of these | In fact, the rains and the timing of seasons, you wouldn't expect to see cases after June even after the vaccination. Also, efficacy was higher in these studies because they had older populations. The efficacy of single-dose strategy in Bangladesh under 5 is less than 58% so we may need to revisit this recommendation | Oral Cholera Vaccinations |
| Gona PN 2020 [40] | Examined the coverage of ORS available in households in Zambia, Malawi and Zimbabwe amongst households with children with diarrhoea using cross-sectional comparative analysis of two demographic health survey cycles | Tri-county cross-sectional survey across 2 time periods and compared DHS data and household questionnaires | Country-wide assessment over two DHS cycles. Plateaued ORS coverage. Lower in rural provinces (Muchinga, Northern, and Central had less than 60% coverage in 2013) | Identified hotspots with lower coverage, also mothers with less education, older or HIV negative had less routine ORS usage despite increase in diarrhoeal deaths. Noted increased diarrhoea expected with climate change effects | Policies needed to strengthen access to appropriate treatments and promote ORS use to be implemented which could help reduce routine deaths from diarrhoea and conversely community deaths from cholera | Case Management |
| Irenge LM 2020 [37] | Genomic analysis of pathogenic isolates of vibrio cholera from eastern DRC (2014–2017) | Lab-based descriptive study | 97 patient isolates from 3 sites in DRC | Phenotypic analysis and WGS for strains from DRC to determine relatedness from DRC and potential to spread to Zed (ST%15 Clade spread here from DRC | Need for enhanced cross-border surveillance | Laboratory/Surveillance |
| Mutale L 2020 [23] | Risk and Protective factors for cholera deaths during an urban outbreak in Lusaka 2017–2018 | Case-control study, administered questionnaire and used univariate logistic regression to calculate matched odds ratios for death | Lusaka between October 2017 and January 2018 and compared 38 decedents and 76 survivors | Mean age was 38 for deaths and 25 for survivors. Odds of death above age 55 was 6.3 with 95% CI:1.2-63.0 or those who did not complete primary school (mOR 8.6, 95%CI:1.8-81.7) | Higher odds of dying with increased age above 55 years or illiterate hence messaging should address these groups and not only traditional print media. Also, need for emphasis on ORS at home as cornerstone of early treatment | Case Management |

*(Continued)*

| First Author | Aims | Study Design | Population and Location | Identified Risk Factor | Possible Mitigative Factor | Theme |
|---|---|---|---|---|---|---|
| Mwape K 2020 [34] | Characterisation of V. cholerae isolates from 2009, 2010, and 2016 outbreaks in Lusaka Province | Lab-based descriptive cross-sectional study that examined 83 isolates from 3 different outbreaks (2009, 2919 and 2016) | Stool and rectal swabs from stored samples in the Lusaka outbreak were examined | Showed high genetic diversity amongst the strains suggesting not only a common source but also rising multidrug resistance. 90% were sensitive to cotrimoxazole, which is different from sensitivities seen in the 2023/2024 outbreak. Ogawa strains were responsible for 2009 and 2016, but Inaba for 2010 | Recommended close monitoring of the V. cholerae strains causing outbreaks due to increasing MDR strains, and reversion to previously sensitive strains | Laboratory/Surveillance |
| Nanzaluka FH 2020 [24] | Risk factors for epidemic Cholera in Lusaka, Zambia - 2017 | Case-control study with controls as neighbours with no diarrhoea during the period. 2:1. tested FRC and the presence of soap in the home | Lusaka, 82 cases and 132 controls in Lusaka District in Dec 2017 | Inadequate supply of safe water by utility companies, i.e., use of shallow wells, private boreholes or water kiosks, contamination of piped water sources (57% of cases and 52% of controls used shallow wells hence resorting to shallow wells, especially during rationing. 84% of cases and 88% of controlled reported inadequate water in their homes from any source | Borehole water was one of the risks, male, close contact of cholera case. All households reported inadequate access to water due to intermittent supply. Need for enhanced investment in municipal infrastructure for centralised water delivery in adequate quantities | Laboratory/Surveillance |
| Reaver S. 2020 [28] | Evaluated the quality and provision of drinking water in six low-income peri-urban communities of Lusaka, Zambia | Examined water samples from 77 unique sites in the 6 communities with matched GPS coordinates at 4 time points between June 2013 and June 2019 | Peri-urban slums in Lusaka - particularly Chaisa, Chazanga, Chipata, Garden, Ngombe and Kanyama. Covering a total population of over 1 million people. They sampled 16 Water Trust boreholes, 23 kiosks linked to Water Trust boreholes, 27 shallow hand-dug wells and 11 privately owned boreholes over the 6 years | These peri-urban communities overlay crystallite dolomite and dolomitic limestone formations that render them extremely vulnerable to groundwater. contamination. Water trusts are private boreholes that treat and provide water to the communities as a supplement to municipal water utility companies. Shallow wells were found to be most contaminated with *E. coli* and the Trusts offered a safer alternative. However, all showed evidence of nitrite contamination (72% for the shallow wells, 25% private boreholes and 16% water trusts) showing vulnerability to faecal contamination even at those depths | Water Trusts provide a safer alternative to underserved populations, however, only cater for 60% of the population who would still need shallow wells. There would be a need to expand the distribution capacity of the trusts and subsidise costs to the population, particularly vulnerable to increase access to safe water, as a bridge to longer-term investment. Added need to expand monitoring of water quality by the Ministry of Health | Water, Sanitation and Hygiene |

*(Continued)*

**Table 2.** (Continued)

| First Author | Aims | Study Design | Population and Location | Identified Risk Factor | Possible Mitigative Factor | Theme |
|---|---|---|---|---|---|---|
| Mwaba J 2020 [19] | Identification of cholera hotspots in Zambia: A spatio-temporal analysis of cholera data from 2008 - 2017 | Descriptive analysis of the cholera outbreaks in the 10 provinces of Zambia based on the data collected from the MOH surveillance platform over 10 years, with additional information from the Demographic Health Survey. Then used Poisson-based space-time scan statistics to estimate spatial districts and hotspots | Cases were noted by district and age and showed 72 of 116 districts had reported cholera cases. 29,080 cases were reported in Lusaka in the 10 years (i.e., 89% of the total 34,950 cases during the period). Limitation was they excluded children under 2 years even in hotspots | Wards in Lusaka that housed high-density communities (only 3 of the 33 wards had the highest risk) Outside of Lusaka, districts that had proximity to water bodies, and movement between neighbouring districts. Hypothesised increase of cases associated with rainfall and flooding. Limited by the lack of access to WASH data | Targeted interventions in the hotspot districts and the particularly affected wards. Recommended for real-time case investigation with GIS mapping for future outbreaks for real-time interventions. This was done during the 2023/2024 outbreak | Laboratory/Surveillance |
| Luchen CC 2021 [55] | Effect of HIV Status and retinol on immunogenicity to OCV in adult population living in an endemic area of Lukanga Swamps, Zambia | Nested study in a cohort of patients in Lukanga swamps followed up for 4 years investigating long-term immunogenicity of OCV | Compared 47 participants and found 24 who were HIV positive | Reduced immunogenicity from HIV positive in line with the CD4 and viral load and so more work is needed | Host factors such as HIV status need to be specifically studied to understand vaccine efficacy and transmission dynamics | Oral Cholera Vaccinations |
| Mwaba J 2020 [32] | Three transmission events of vibrio cholerae 01 into Lusaka Zambia | Examination of 72 VC isolates from the 3 different outbreaks to compare the multilocus variable number tandem repeat analysis (MLVA) and whole genomic sequencing | Isolates from stored stool samples from the Lusaka outbreaks, and Mpulungu and Chiengi for the later outbreaks. Mpulungu isolates identical to Lusaka | MLVA of isolates from the 2009,2016 and 2017 outbreaks shows that 3 separate transmission events occurred. Isolates from 2016 and 2017 in Kanyama were distinct and showed the vulnerability of these wards | Dispute the endemicity theory since the isolates were genetically distinct even in concurrent years. Instead, advocate for measures to prevent reintroduction and recurrent spread of vibrio into Zambia | Laboratory/Surveillance |
| Mwaba J 2021 [51] | Serum vibriocidal responses when second doses of oral cholera vaccine are delayed 6 months in Zambia | Open-label phase 2 RCT in healthy adults to compare vaccine vibriocidal GMT at 2 weeks and 6 months (so 14 days after OCV was given) | 152 Adults in Lukanga Swamps (70 km from Kabwe) dosed between October 2017 and April 2018 | People residing in fishing camps, with high mobility to Lusaka where an outbreak was happening. Hence the delayed second dose is acceptable | No difference hence suggesting the flexible dosing for the 2d dose is acceptable | Oral Cholera Vaccinations |
| Sack D 2021 [36] | Contrasting epidemiology of Cholera in Bangladesh and Africa | Review article comparing patterns of cholera outbreaks in Bangladesh and in Cameroon, at sentinel sites, the team had set up as part of enhanced surveillance efforts | Compared outbreaks in Bangladesh to what was seen in Cameroon, DRC, Zambia, Zanzibar and Uganda | Ganges Delta is seasonal, but in Africa inconsistent with explosive outbreaks. Elimination of lineages and reintroduction possibly by travellers. Need to re-examine the use of OCV and WASH. Proposed that reintroduction more likely than environmental reservoirs which spring up when climatic conditions are favourable since the lineages are different across the years. Evident association with climatic factors influencing outbreaks in Africa | Need for improved surveillance systems to ensure that estimated burden of cholera in Africa is not an overestimation. Reintroduction events also warrant a need for better cross-border collaboration. Champion for elimination efforts to be done at district level and prevent reintroduction. Need to explore household transmission dynamics and effects, and further studies to explore environmental reservoirs | Laboratory/Surveillance |

*(Continued)*

**Table 2.** (Continued)

| First Author | Aims | Study Design | Population and Location | Identified Risk Factor | Possible Mitigative Factor | Theme |
|---|---|---|---|---|---|---|
| Malata M 2021 [16] | Quantitative exposure assessment to Vibrio cholerae through consumption of fresh fish in Lusaka province | Simulation study using Swift Quantitative Microbial Risk Assessment (sQMRA) model framework following pathogen numbers in 3 transmission settings | Used secondary data from MOH sources, then conducted household questionnaires in different social strata in Lusaka | Low risk of cholera acquired through consumption across different pathways in Lusaka because of the food preparation practises here (raw fish rarely eaten) | Messaging should be clear and need not mention fish as a transmission route as this has been disproved | Laboratory/Surveillance |
| Colston J 2022 [63] | Associations between eight earth observation-derived climate variables and enteropathogen infection: an independent participant data metanalysis of surveillance sites with Broad spectrum nucleic acid diagnostics | Metanalysis of studies from different countries bringing together data sources from molecular and climatic zones | 64,788 eligible stool samples from 20,760 children were analysed | Rotavirus infection decreased markedly following increasing 7-day average temperatures- a relative risk of 0.76 (95% confidence interval: 0.69-0.85) above 28°C-while ETEC risk increased by almost half, 1.43 (1.36-1.50), in the 20–35°C range. Risk for all pathogens was highest following soil moistures in the upper range. Humidity was associated with increases in bacterial infections and decreases in most viral infections | supports evidence of a U-shaped association between rainfall and enteric pathogen proliferation due to concentration-dilution hypothesis greater precipitation variability due to climate change on diarrhoea-causing pathogens is not certain and is likely to be highly species and location-specific | Water, Sanitation and Hygiene |
| Fakoya B 2022 [60] | Non-toxigenic vibrio cholera challenge strains for evaluation vaccine efficacy and inferring mechanisms of protection | Preclinical studies on Zchol strains that have been shown to effectively induce immunity | Infant mice models to show toxicity vs immunity garnered from the Zchol strains | Created a non-toxigenic Zchol strain that can be used in controlled human infection studies. However, noted that these human challenge studies are often not done in endemic countries. But proposed that such a strain could be useful to one day | Recommended for additional research in endemic areas such as ours to test out new vaccines, plus other studies into transmission dynamics | Oral Cholera Vaccinations |
| Chota P 20222 [65] | From The Plague Horrors of Cholera, What Partnership Lessons Can Be Learnt in Addressing COVID-19 in Zambia | Qualitative approach. 26 community leaders and health care professionals. Bergen model of collaborative functioning to guide data analysis | Chipata compound Lusaka, reviewing the response to the 2017–2018 cholera outbreak | Challenges in the partnership collaboration included inadequate resources, poor communication, poor coordination, lack of clear shared vision, reactive response, poor involvement of the community, hegemonic powers and mistrust | "Despite the attempts at co-ordination, ministries have a tendency of operating in isolation, this has resulted in lack of a clear shared vision. This also contributes to duplications of tasks in trying to prevent an outbreak of cholera." | Leadership & Coordination |

*(Continued)*

**Table 2.** (Continued)

| First Author | Aims | Study Design | Population and Location | Identified Risk Factor | Possible Mitigative Factor | Theme |
|---|---|---|---|---|---|---|
| Meki CD 2022 [43] | Community Level interventions for mitigating the Risk of water-borne diarrhoeal disease - a systematic review | Systematic review of full papers published across the world between 2009 and 2020 describing various community-level interventions that could reduce diarrhoeal disease | Worldwide, including publications from Zambia | Poor WASH and poor healthcare systems were identified as risks for cholera outbreaks, minimal evidence of the efficacy of vaccines, and the need for improved surveillance | Recommend that interventions for waterborne diseases be concentrated in developing countries as they are the main areas where these diseases are most common. The interventions must also concentrate mostly on control of the disease in children even though adults are also affected. At a community level, vaccines seem to be the most effective interventions and are probably the easiest to implement | Oral Cholera Vaccinations |
| Ng'ombe H 2022 [53] | Immunogenicity and waning immunity from the oral cholera vaccine (Shanchol™) in adults residing in Lukanga Swamps of Zambia | Sub study of the nested control trail in Lukanga swamps | Cohort of 223 patients aged 18–65 | Seroconversion was only 25% for organ and inaba after 1 dose. Waned below baseline by 12 months and increased at 36 months maybe natural exposure so that would be a good time to revaccinate | Vibriocidal Antibodies wane by month 36 hence recommendation for repeat vaccination campaigns every 3 years | Oral Cholera Vaccinations |
| Sialubanje C 2022 [47] | Effectiveness of two doses of Euvichol plus oral cholera vaccine in response to the 2017/2018 outbreak: a matched case-control study in Lusaka Zambia | Matched case-control study following mass vaccination campaign in 2018 in Lusaka | 79 cases and 316 controls identified from 5715 patients who had been recorded at any of the 6 CTCs in Lusaka | Conditional logistical regression analysis showed a significant association between two doses of the Euvichol-plus OCV and vaccine protection (AOR = 0.19; 95% CI 0.16 to 0.28) with vaccine effectiveness of 81% (95% CI 72.0% to 84.0%; p value <0.01) (Table 2). The effectiveness of any (one or more) doses of Euvichol- plus vaccine was 74%. It was the first use of Euvichol here and suggested that the two-dose strategy was better than a single dose in outbreak setting' | Recommended for more longitudinal studies to determine the long-term effectiveness of two doses of OCV among the vaccinated populations in the local context. Further research is also required to determine the effectiveness and usefulness of Euvichol-plus vaccine in conferring herd immunity among non-vaccinated individuals during mass immunisation and to determine the required minimum coverage. Finally, further research is needed to determine Euvichol-plus vaccine effectiveness among people living with HIV and its usefulness among these populations | Oral Cholera Vaccinations |
| Chisenga CC 2023 [54] | Assessment of the influence of ABO blood groups on OCV immunogenicity in a cholera endemic area in Zambia | Longitudinal study nested in the clinical trial in Lukanga Swamps with patients being followed up 4 years post-vaccination. Measured GMT at day 28, 6M, 12M, 24M, 30M, 36M and 48 M | Lukanga Swamps, 4-year cohort, 133 patients included in the assessment | Sub-study of their 4-year cohort found no influence of ABO on the influence of vaccine uptake and cholera response | No support for ABO influencing vaccine uptake, but an important study opening the gateway to investigate host-specific factors associated with cholera acquisition | Oral Cholera Vaccinations |

*(Continued)*

**Table 2.** (Continued)

| First Author | Aims | Study Design | Population and Location | Identified Risk Factor | Possible Mitigative Factor | Theme |
|---|---|---|---|---|---|---|
| Maity B 2023 [31] | Model-Based estimation to expected time to cholera extinction in Lusaka, Zambia | Exploration of epidemiological modes of transmission. Used weekly case numbers and inputted them into two transmission modes human-to-human vs environment-to-human | Mathematical modelling for Lusaka based on cases between October 2017 and May 2018 | Calculating R0 both modes were active in first wave but Environment-to-human route was a dominant mode in second wave - heavy rainfalls, floods and reducing in water and sanitation led to an indirect mode of transmission due to increased environmental vibrio. Time to extinction was calculated as Cholera can last 8 months - 6.5 years. they quoted the MCEP as supporting their findings | WASH interventions and Mass vaccinations should be combined to end cholera | Water, Sanitation and Hygiene |
| Mukonka VM 2023 [48] | Euchivol-plus vaccine campaign coverage during the 2017/2018 cholera outbreak in Lusaka district; a cross-sectional descriptive study | Descriptive cross-sectional analysis of OCV coverage in 2017/2018 outbreaks using satellite map-based sampling to identify households | 1691 participants from four localities in Lusaka (Kanyama, Chawama, Chipata and Matero) | Reported OCV administratively was much higher than actual coverage, with only 66% getting two doses and, an 18% dropout rate. Majority vaccinated were female, could that explain our male predominance now? Reliance on administration has always been lower because of data inaccuracies and also studies to look into reasons for high vaccine dropout rates | Recommend interventions during OCV campaigns that target particular patient groups (men in this case) and risk communication initiatives to reduce dropout rates | Oral Cholera Vaccinations |
| Gething W 2023 [14] | Geospatial analysis of cholera risk in Lusaka to inform improved water and sanitation provision following 2018 outbreak | Conducted geospatial mapping of the cases and their communities to produce granular risk maps followed by mathematical modelling of risk factors against different scenarios to predict reductions in cholera cases based on the different proposed interventions | Lusaka wide | Risk factors here increased density, unimproved sanitation, high sanitation index and prevalence of E. coli contamination in water sources. Plus decreased risk with distance from flooding | Provision of flush-to-sewer to all households reduced 90% of cholera cases if implemented. Next was provision of piped water to all households would reduce by 61%. Proposed interventions would need to be done at ward level to counter high-cost implications | Water, Sanitation and Hygiene |

*(Continued)*

**Table 2.** (Continued)

| First Author | Aims | Study Design | Population and Location | Identified Risk Factor | Possible Mitigative Factor | Theme |
|---|---|---|---|---|---|---|
| Wiens KE 2023 [21] | Systematic review estimating the proportion of clinically suspected cholera cases that are true vibrio cholerae infections - a systematic review and metanalysis | Meta-analysis of 119 papers from 30 countries | Worldwide meta-analysis of 30 countries over 20 years from 2000 to 2023 | Suggested that the number of cases meeting the case definition may be higher than true cholera cases especially outside outbreak seasons, so we needed more specific testing (only 52% representing true V. cholerae) outside of outbreak seasons | Need to improve clinical cholera surveillance (e.g., patients visiting traditional healers and pharmacists) may help understand the true burden especially early in outbreaks - i.e., recommend a clinical early warning system | Laboratory/ Surveillance |
| Chanda TC 2024 [66] | Understanding and addressing the cholera outbreak in Zambian communities | Mixed method qualititative and quantitative study of the people in MOH and district staff plus frontliners | The sample size involved a total of 42 respondents which included two (2) officials from Ministry of health, four (4) medical doctors, one coming from each selected hospital. Sixteen (16) Nurses, four coming from each selected hospital. Twenty (20) support staff, five coming from each selected hospital in communities A, B, C, and D. | Cholera persists due to weak healthcare infrastructure, poor sanitation, inadequate access to clean water, and limited community awareness. Socioeconomic hardship hinders adoption of preventive measures, while climate change—through floods and droughts—exacerbates outbreak severity and frequency, particularly in vulnerable communities with limited capacity to respond effectively. | Strengthen health infrastructure and update emergency plans. Improve water, sanitation, and hygiene (WASH) services in high-risk areas. Launch targeted awareness campaigns, implement poverty alleviation programmes, and promote community ownership of health. Integrate climate adaptation into public health strategies to reduce cholera risks and enhance long-term outbreak resilience. | Leadership and Coordination |
| Chisenga C 2024 [59] | Examination of seroconversion and kinetics of vibriocidal antibodies during the first 90 days of revaccination with OCV in an endemic population. | A prospective study following a cohort of patients who had been vaccinated 4 years prior vs naïve. Bloods collected at 5 time points and vibriocidal antibodies compared for an estimate of the protective immunity provided | Lukanga Swaps in Kapiri Mposhi, 182 in the final analysis | Seroconversion was similar regardless of previous vaccination status with rapid waning | Proposed revaccination at day 30 as the antibodies are higher than baseline in naïve individuals | Oral Cholera Vaccinations |

*(Continued)*

**Table 2.** (Continued)

| First Author | Aims | Study Design | Population and Location | Identified Risk Factor | Possible Mitigative Factor | Theme |
|---|---|---|---|---|---|---|
| Libanda B 2024 [62] | Recent and future exposure of water, sanitation and hygiene systems to climate-related hazards in Zambia | Utilised the WHO/UNICEF Joint Monitoring Programme on Water, supply, Sanitation and Hygiene for 2000–2021 to estimate WASH coverage in Zambia. Then combined with the Global Precipitation Climatology Centre's monthly precipitation data, they conducted simulations from the latest Coupled Model Intercomparing Project Phase 6 (CMIP6) | Nationwide | Nationally, 65% of the population have access to safe drinking water, 32% rely on unimproved water services, 6% on limited service and 7% depend on surface water. 32% of the population have access to basic sanitation, 20% limited sanitation, 11% depend on open defecation, and 37% use unimproved sanitation - with urban dwellers, particularly in the peri-urban slums that represent 70% of the population, fairing worse than rural ones in terms of sanitation. Less than half the population have hygiene services. Association with rainy season 71% of cholera outbreaks, a 300% increase with El Nino. Drought is the most common high-impact hazard, and drought-driven water shortages will lead more people to unsafe water sources. Besides increasing people's exposure to contaminated water, these climatic events also lead to deterioration of sanitary services | Need to build urban resilience against flooding. All drainages need to be interconnected, particularly in Lusaka which has peri-urban settlements. Climate resilience should lead to ongoing WASH investment, particularly when seeking to secure groundwater sources which will drop to critically low levels during the drought years projected in the middle of the century. Rural populations facing drought may migrate more to peri-urban settlements, further compounding the water-stressed situations and the likelihood of cholera outbreaks | Water, Sanitation and Hygiene |
| Mbewe N 2024 [7] | Survival analysis of patients with cholera admitted to treatment centres in Lusaka, Zambia | Cohort analysis of in-patient data in Lusaka during the 2023/204 outbreak | 1529 patient survival outcomes described between 10th January and 30th April 2024 admitted in any CTC in Lusaka | Older age and the presence of comorbid conditions are associated with higher odds of mortality during admission. Previous vaccination seen as protective | Need to understand better the interplay of comorbidities on case management, particularly the implication on fluid management | Case Management |
| Kateule E 2024 [6] | Multisectoral approach for the control of cholera outbreak - lessons and challenges from Lusaka district - Zambia, October 2023 - February 2024 | Descriptive observational study f cholera response activities in Lusaka district. Mixed method qualitative and quantitative with review of surveillance records and documented response interventions and challenges using situation reports | Lusaka District with 13122 cases and 498 deaths as at 12th February 2024 | Despite having a well-established system for coordinating technical support and resource mobilization, inadequate sanitation and limited access to clean water remained potential risks for cholera outbreaks in Lusaka district. | A multisectoral coordination for improved sanitary systems, access to clean water, health education strategies, and vaccination campaigns contributed to the decline in cholera cases. | Leadership & Coordination |

*(Continued)*

**Table 2.** (Continued)

| First Author | Aims | Study Design | Population and Location | Identified Risk Factor | Possible Mitigative Factor | Theme |
|---|---|---|---|---|---|---|
| Ng'ombe H 2024 [42] | Comparative analysis of cholera serum vibriocidal antibodies from Convalescent and vaccinated adults in Zambia | Comparative analysis of cohort of patients who had received OCV and those with natural immunity. Vibriocidal antibodies plotted as geometric mean titres in naturally infected vs vaccinated individuals | 50 from Eastern province and 56 from Central province | Delays in vaccination booster doses due to global shortages | Two dose vaccination superior to single dose with lowest titers at day 0–9 vs peak at day 10–19. Recommend post infection vaccination after 40 days for sustained immunity and prolonged protection | OCV |
| Xiao S 2024 [35] | New Vibrio cholerae sequences from Eastern and Southern Africa alter our understanding of regional cholera transmission (preprint) | 114 high quality V. cholerae genomes combined with 1385 previously published genomes to conduct phylogenetic and other analyses used to better understand cholera transmission and circulation in Southeastern Africa. | 114 V. cholerae O1 genomes from samples collected in Kenya, Tanzania, Uganda, Malawi and Zambia from 2007-2019 | frequent co-circulation of multiple combinations of lineages. These findings also emphasise the importance of a regional approach to cholera surveillance and 22, as outbreaks in neighbouring countries are connected both temporally (e.g., spikes in cases occur around the same time; and molecularly (e.g., sequences from V. cholerae in multiple countries are highly related) | Cholera containment and mitigation, which may require cooperation across country borders. | Lab/ Surveillance |

communities was also posited as an avenue to improve acceptance of public health interventions such as vaccines [77]. In parallel, we advocate for strengthened global collaboration in medical education and the bi-directional exchange of knowledge between low- and high-income countries. Such partnerships can enhance local research capacity, foster contextual innovation, and accelerate regional vaccine manufacturing. This aligns with recommendations from the President of the Republic of Zambia, in his role as the WHO Global and Southern Africa Development Community (SADC) Regional Cholera Control Champion, who has called for the development of regional vaccine production hubs to improve timely access and health security across the Global South [78].

Evidence for patient-specific case management modalities using host genomics is nascent. Research into the host microbiome is in early phases with mixed results but gives potential for newer treatment modalities such as probiotics and phage therapy against Vibrio cholerae [79,80]. Recent studies have highlighted the complex and sometimes contradictory role of the gut microbiome in cholera susceptibility and transmission [81–83]. Certain commensal bacteria have been associated with protective effects, potentially by competing with *Vibrio cholerae* for nutrients or attachment sites in the intestinal mucosa [81]. Conversely, disruptions of the gut microbiota—due to factors such as malnutrition, prior antibiotic use, or environmental exposures—may reduce colonization resistance, thereby increasing an individual's vulnerability to infection [82]. Moreover, variability in microbiome composition across populations and geographic regions may partly explain differences in outbreak dynamics and individual disease severity [83]. These findings underscore the importance of considering host–microbe interactions in cholera prevention strategies. Notably, emerging evidence suggests that baseline gut microbiota composition may influence oral cholera vaccine efficacy, particularly in low-income settings, prompting further investigation into microbiome-based correlates of protection [58]. Additionally, bacteriophages and probiotics are being explored as adjunctive therapies to enhance cholera case management by modulating gut flora and directly targeting *V. cholerae*, though these approaches remain under active research and are not yet standard practice [82–84].

Isolates from the 2023/2024 outbreak in Zambia have yet to be fully analysed for host and pathogen genomics. However, recent sequencing results from Malawi give insight into a possible new transmission event into the subcontinent, which bears a close resemblance to strains of Asian origin [85]. Zambia and Malawi share many porous borders, and trade and intermarriage are common between the people. The Malawian study postulated that the strain of Vibrio in their 2022/2023 outbreak, the worst in Malawian history, was a highly successful cone of pandemic potential worsened by humanitarian and climate crises and then propagated by suitable environmental factors [85]. This agrees with earlier findings suggesting that outbreaks in Kanyama and other hotspots like the fishing villages, were due to a combination of recent introduction of newer pathogenic strains and favourable environmental factors like deplorable WASH status [32,34,35]. This also underscores the importance of joint cross-border surveillance and response activities in the region [6,29,35–37].

Challenges and gaps persist in cholera elimination efforts in Zambia. The need for a multisectoral, decentralised approach is evident, as no single intervention would remove all the various identified risk factors. The studies reviewed showcased different aspects of interventions during outbreak settings or vaccination efforts in a reactive response. What can be seen is that cholera outbreaks in Zambia and Africa as a whole are progressively larger [6,25,77] and call for enhanced multisectoral and cross-border collaboration [65,77]. Without environmental source control, such as improving flush-to-sewage plumbing systems and overall climate-resilient solutions, it can be anticipated that the number of outbreaks in the region will continue to increase [65,66,76,77]. Our findings broadly confirm the need to align health and WASH investments with the GTFCC's Roadmap to Cholera Elimination by 2030 [1] but also highlight the need for additional research across the various pillars to ensure tailored solutions are adaptable to the local setting and able to inform best practice.

While the study is primarily based on a scoping review methodology, limiting the application of statistical tests and resulting in largely descriptive recommendations, there are several notable strengths. The review synthesises a broad and complex body of evidence on cholera control in Zambia, offering a consolidated narrative that clearly outlines the key challenges and persistent gaps in the country's elimination efforts. By mapping these barriers against existing interventions, the study provides practical insights that can inform more targeted and strategic planning for the next iteration of the National Cholera Control Plan.

Importantly, the review identifies priority areas for future research, including best approaches for implementing community-centric surveillance and CATI. It also highlights critical gaps in patient-level data on survival outcomes and transmissibility, particularly in vulnerable populations such as the elderly and pregnant women. There is a clear need for studies exploring the influence of co-morbidities, host genetic factors (e.g., gut microbiome), and household-level dynamics on disease progression and spread. The potential of metagenomic technologies for enhancing point-of-care testing and linking surveillance to clinical outcomes is underscored as an emerging frontier. Similarly, the role of adjuvant therapies in vaccination and treatment regimens remains underexplored. Lastly, the review emphasises the growing importance of understanding how climate change, through both drought and flooding, affects WASH infrastructure and health outcomes. These insights contribute meaningfully to the global evidence base and offer direction for researchers and policymakers working toward the 2030 cholera elimination goal. In particular, the paucity of peer-reviewed literature on community engagement in cholera control in Zambia, as shown in our review (Fig 2), points to a critical gap in evidence. We highlight the need for more implementation research to identify effective, scalable models for community engagement in cholera surveillance, vaccination uptake, and WASH interventions. Strengthening this evidence base is essential for designing context-specific strategies that are both sustainable and responsive to the needs of high-risk communities.

## Conclusion

This scoping review collated evidence supporting a decentralised approach to cholera control in Zambia and Sub-Saharan Africa overall. Two key findings emerge from the analysis: first is the steady increase in cases and deaths over the years,

despite adopting the first iteration of the Multisectoral Cholera Elimination Plan in 2019, and an anticipated increase in the coming years with rapid population growth and changing climate. The second key finding is that a wealth of evidence has already been generated in Zambia regarding best practices towards cholera control. There is a continued need to advocate strongly for multisectoral interventions with an alignment of health and WASH investment at the district and ward level, to align with this decentralised approach. The findings suggest multiple areas of further research considering the endemicity of cholera in Zambia. We propose that our insights and recommendations can inform policymakers in crafting guidelines for implementing ward-level interventions, and these will be integrated into the next iteration of the National Cholera Control Plan. We hope that the lessons from here can be applied in other sub-Saharan African countries facing similar challenges and seeking to internalise the Global Roadmap for Cholera Control by 2030.

## Supporting information

**S1 Table. Preferred Reporting Items for Systematic reviews and Meta-Analyses extension for Scoping Reviews (PRISMA-ScR) Checklist.** Legend: This table outlines the key reporting elements recommended for scoping reviews to ensure transparency, methodological rigour, and reproducibility. Each item corresponds to a section of the review and indicates whether it has been addressed in the manuscript.
(DOCX)

## Acknowledgments

This work is part of ongoing efforts from the Zambia National Public Health Institute, as the Secretariat of the National Cholera Control Taskforce, to better understand efforts towards Cholera Control in the Region. Many thanks to various task force members and partners who were directly and indirectly involved in this work.

## Author contributions

**Conceptualization:** Nyuma Mbewe, John Tembo, Roma Chilengi, Nathan Kapata, Martin Peter Grobusch.

**Data curation:** Nyuma Mbewe, John Tembo, William Ngosa, Nathan Kapata.

**Formal analysis:** Nyuma Mbewe, Martin Peter Grobusch.

**Investigation:** Nyuma Mbewe, William Ngosa, Martin Peter Grobusch.

**Methodology:** Nyuma Mbewe, John Tembo, Nathan Kapata, Martin Peter Grobusch.

**Project administration:** Nyuma Mbewe.

**Supervision:** Mpanga Kasonde, Kelvin Mwangilwa, Paul Msanzya Zulu, Joseph Adive Sereki, Kennedy Lishipmi, Lloyd Mulenga, Roma Chilengi, Nathan Kapata, Martin Peter Grobusch.

**Validation:** Mpanga Kasonde, Kelvin Mwangilwa, Paul Msanzya Zulu, Joseph Adive Sereki, William Ngosa, Kennedy Lishipmi, Lloyd Mulenga, Roma Chilengi, Nathan Kapata, Martin Peter Grobusch.

**Visualization:** Nyuma Mbewe, Joseph Adive Sereki, Roma Chilengi, Martin Peter Grobusch.

**Writing – original draft:** Nyuma Mbewe, John Tembo, Roma Chilengi, Nathan Kapata, Martin Peter Grobusch.

**Writing – review & editing:** Nyuma Mbewe, John Tembo, Mpanga Kasonde, Kelvin Mwangilwa, Paul Msanzya Zulu, Joseph Adive Sereki, William Ngosa, Kennedy Lishipmi, Lloyd Mulenga, Roma Chilengi, Nathan Kapata, Martin Peter Grobusch.

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
