## [Decision Letter · Decision Letter 0]

PNTD-D-24-01118

Navigating the Cholera Elimination Roadmap in Zambia - a Scoping Review (2013-2023)

Dear Dr. Mbewe,

Thank you for submitting your manuscript to PLOS Neglected Tropical Diseases. After careful consideration, we feel that it has merit but does not fully meet PLOS Neglected Tropical Diseases's publication criteria as it currently stands. Therefore, we invite you to submit a revised version of the manuscript that addresses the points raised during the review process.

Please submit your revised manuscript within 60 days May 24 2025 11:59PM. If you will need more time than this to complete your revisions, please reply to this message or contact the journal office at plosntds@plos.org. Please include the following items when submitting your revised manuscript:

We look forward to receiving your revised manuscript.

Kind regards,

Jeffrey H Withey

Academic Editor

Victoria Brookes

Section Editor

Shaden Kamhawi

co-Editor-in-Chief

Paul Brindley

co-Editor-in-Chief

**Additional Editor Comments (if provided):**

Please respond to each of the reviewers' comments and indicate where changes have been made in the revised manuscript. Please note that one reviewer has comments in the attachment and the other in the body of this email.

In particular, please note that the methods must reflect those of a scoping review, i.e. PRISMA-ScR guidelines, to ensure accuracy and reproducibility of findings.

**Journal Requirements:**

At this stage, the following Authors/Authors require contributions: Nyuma Mbewe. Please ensure that the full contributions of each author are acknowledged in the "Add/Edit/Remove Authors" section of our submission form.

- ® on pages: 16, 17, 20, 30, and 32.

**Reviewers' Comments:**

Reviewer's Responses to Questions

**Key Review Criteria Required for Acceptance?**

**Methods:**

-Are the objectives of the study clearly articulated with a clear testable hypothesis stated?

-Is the study design appropriate to address the stated objectives?

-Is the population clearly described and appropriate for the hypothesis being tested?

-Is the sample size sufficient to ensure adequate power to address the hypothesis being tested?

-Were correct statistical analysis used to support conclusions?

-Are there concerns about ethical or regulatory requirements being met?

Reviewer #1: (No Response)

Reviewer #2: The Methods need to be clarified, coherence with the research questions defined in introduction and the structure of the Results section improved, and further details provided on how data were analysed or synthesised. The PRISMA check-list (for scoping or systematic review) is a useful guide to structure the manuscript - it is mentioned in Results (L. 122) but not adhered to.

The terms "scoping review" and "narrative review" are used interchangeably in the manuscript to refer to the work. However, the research questions defined at L. 97-99 require synthesising evidence on (i) cholera in Zambia and (ii) cholera control strategies, and they would be best addressed through a systematic review (see Munn et al., BMC Medical Research Methodology, 2018).

L. 112: the search terms included "Zambia" and articles not related to Zambia were excluded. This makes the scope of the review very specific to the Zambian context and prevents identification of relevant evidence from similar settings to address the research question.

L. 109 / 118: what is the point of mentioning Embase if this database could not be accessed? Consider moving to discussion / limitations rather than keeping this in Methods and Results.

**Results:**

-Does the analysis presented match the analysis plan?

-Are the results clearly and completely presented?

-Are the figures (Tables, Images) of sufficient quality for clarity?

Reviewer #1: (No Response)

Reviewer #2: The Results section summarises findings from studies included in the review but does not include any assessment of the strength of evidence. This is related to my comment above; a systematic review with a risk-of-bias assessment would be more appropriate approach to synthesise evidence and formulate recommendations on that basis.

L. 111: what was done with the 11,300 records from Google Schholar and why are they not reflected in Fig 1? It is difficult to reconcile the manuscript text with Fig 1, as several numbers do not match.

L. 123-126: this belongs to Methods.

L. 147-148: only two papers in Table 1 appear to include more than one year of cholera surveillance and support this statement, and there is likely an issue with the reference number (Table 1 vs. text).

L. 148-152: this sentence is unclear, especially "...the proportion of clinically suspected cholera cases and not all confirmed by culture..." - are many cases confirmed by culture outside clinical settings?

L. 187, 189: I believe 'Vibrio' is used instead of 'vibriones/vibrioni' in Englishs - to be checked.

L. 213-217: how many studies is this estimate of 45% facility vs. 55% community deaths based on, and which time period does it refer to? Include references.

L. 223: "antimicrobial resistance" is vague - does it have any implications for patient management / treatment?

L. 235: don't you mean that reported administrative coverage is higher than actual coverage?

**Conclusions:**

-Are the conclusions supported by the data presented?

-Are the limitations of analysis clearly described?

-Do the authors discuss how these data can be helpful to advance our understanding of the topic under study?

-Is public health relevance addressed?

Reviewer #1: (No Response)

Reviewer #2: It is not entirely clear how the Discussion and Conclusions are supported by the Results. For instance:

- The statement at L. 314-316 may be sensible but how it is supported by evidence (from the Results section) on the effectiveness of decentralised surveillance and case-area targeted interventions

- Fig 2 shows that there are very few publications on Community Engagement - but this is not reflected upon in Discussion (for identifying future research needs)

**Editorial and Data Presentation Modifications?**

Reviewer #1: (No Response)

Reviewer #2: See above.

**Summary and General Comments:**

Reviewer #1: (No Response)

Reviewer #2: This review is relevant in the context of cholera resurgence in Zambia and across the continent. However, a systematic review including an assessment of the strength of evidence related to the effectiveness of different intervention strategies would more appropriate to address the research questions and inform the development of a sound strategy for cholera elimination. The inclusion from other settings outside Zambia should also be considered to strengthen the evidence synthesis and inform the prioritisation of interventions. I have provided more detailed comments under the respective manuscript sections.

Details

L. 68: there is an issue with the citation

L. 74: the abbreviation WASH is already defined at the beginning of this paragraph

Fig. 2: typo

PLOS authors have the option to publish the peer review history of their article (what does this mean? ). If published, this will include your full peer review and any attached files.

**Do you want your identity to be public for this peer review?** For information about this choice, including consent withdrawal, please see our Privacy Policy .

Reviewer #1: **Yes: ** Md Taiufiqul Islam

Reviewer #2: No

**Figure resubmission:**
---

## [Editor Report · Decision Letter 1]

Dear Dr Mbewe,

We are pleased to inform you that your manuscript 'Navigating the Cholera Elimination Roadmap in Zambia - a Scoping Review (2013-2024)' has been provisionally accepted for publication in PLOS Neglected Tropical Diseases.

Best regards,

Jeffrey H Withey

Academic Editor

Victoria Brookes

Section Editor

Shaden Kamhawi

co-Editor-in-Chief

Paul Brindley

co-Editor-in-Chief

---

## [Editor Report · Acceptance letter]

Dear Dr Mbewe,

We are delighted to inform you that your manuscript, "Navigating the Cholera Elimination Roadmap in Zambia - a Scoping Review (2013-2024)," has been formally accepted for publication in PLOS Neglected Tropical Diseases.

Best regards,

Shaden Kamhawi

co-Editor-in-Chief

Paul Brindley

co-Editor-in-Chief
